# TIMBER: TRAINING-FREE INSTRUCT MODEL REFINING WITH BASE MODEL VIA EFFECTIVE RANK

## ABSTRACT

Post-training, which elicits a pretrained Base model into the corresponding Instruct model, is widely considered to be superficial. In this work, we first reinforce this hypothesis by providing novel quantitative evidence from the weight level that the effective rank (eRank) remains negligibly changed. However, this superficiality also suffers a critical trade-off, improving the exploitation capabilities at the cost of limiting its exploration. To tackle this issue, we propose Timber, a simple yet effective *training-free* method that enhances the exploration capability of the Instruct model while preserving its exploitation. The key insight is to partially revert Instruct towards the paired Base model by subtle yet targeted refinement of the weight deltas. Extensive experiments on Llama and Qwen series demonstrate that Timber consistently improves vanilla Instruct models, particularly on Pass@k performance. Our findings offer new insights into the post-training stage at the weight level and practical strategies to refine the Instruct model *without* training.

## 1 INTRODUCTION

Large Language Models (LLMs), such as Qwen3 (Yang et al., 2025), Llama 3 (Grattafiori et al., 2024), and Deepseek R1 (Guo et al., 2025), have achieved superior success in Natural Language Processing (NLP), especially in reasoning tasks (Huang & Chang, 2023). To train these LLMs, a Base model is first pretrained on huge amounts of data. After that, a post-training stage is applied to train an Instruct model, adopting supervised finetuning (SFT) and reinforcement learning (RL) to elicit alignment and reasoning ability (Yang et al., 2025). The post-training stage tends to be superficial, i.e., post-training only utilizes the pattern contained in the Base model acquired during pre-training (Yue et al., 2025; Zhou et al., 2023a; Ye et al., 2025; Muennighoff et al., 2025).

In this paper, we investigate the Base and Instruct models through the lens of effective rank (eRank, (Roy & Vetterli, 2007)), providing a novel weight-level perspective on the superficiality of post-training. Specifically, eRank quantifies the effective dimensionality of a weight matrix by measuring the uniformity of its singular value distribution, reflecting the intrinsic representational capacity (Schumacher, 1995; Wei et al., 2024). As shown in Figure 1, the eRanks of corresponding linear layers from the Base and Instruct models are almost identical. We can find that post-training induces only negligible changes to the effective dimensionality, offering new supporting evidence from the weight level for its superficiality (Wu et al., 2025a).

However, such superficiality of post-training also suffers a critical trade-off between exploitation and exploration. Specifically, while the Instruct model achieves higher Pass@1 in reasoning tasks, it lags behind on Pass@k for relatively large k (Wang et al., 2025a; Yue et al., 2025; Zhu et al., 2025). In summary, superficial post-training suppresses the sampling space and thereby limits the performance potential. While recent works have sought to mitigate this limitation by introducing additional training objectives (Chen et al., 2025) or external tools (Wang et al., 2025a), these methods invariably incur significant overhead during training or inference.

To this end, we propose **Timber**, a simple yet effective training-free method to enhance an Instruct model using its paired Base model at the weight level. Inspired by the model merge (Yang et al., 2024; Zhang et al., 2024), our key design is to refine the weight delta between the two models, partially reverting the Instruct model towards its Base state to improve exploration ability. Specifically, our method first decomposes the weight delta using SVD and identifies the head and tail components

of the singular values via eRank. Subsequently, the tail components are either removed or attenuated. Through such fine-grained refinement, Timber achieves a better trade-off between exploitation and exploration.

We evaluate the proposed Timber on models including Llama and Qwen series across a suite of popular benchmarks. Experimental results demonstrate that Timber consistently outperforms the vanilla Instruct models, confirming its effectiveness and robustness. Further analysis reveals that this performance gain is largely attributed to a significant improvement in Pass@k scores, which underscores the ability of Timber to enhance the exploration capabilities. Our contributions can be concluded as follows:

- We propose to revisit paired Base and Instruct models via eRank, providing a more granular understanding of post-training superficiality at the weight level.

- We propose Timber, a simple yet effective training-free method that enhances the exploration capability of the Instruct model while preserving its exploitation. The key is to partially revert the Instruct model towards its Base via refining the weight delta.

- We demonstrate the effectiveness and robustness of the proposed Timber via results on various LLMs and benchmarks.

## 2 PRELIMINARY AND ANALYSIS

### 2.1 BACKGROUND

**Superficial Post-training.** Typically, the training of LLM follows a two-stage pipeline (Yang et al., 2025). The first step is to build a Base model with rich knowledge by pretraining on a large amount of training data. After that, we perform post-training on Base to elicit the instruction following and reasoning abilities through the SFT and RL (Shao et al., 2024). Due to the mess of naming, we detailed paired Base and Instruct models in Table 1.

Table 1: Paired Base and Instruct models.

| Base | Instruct | Thinking |
|------|----------|----------|
| Llama-3.1-8B | Llama-3.1-8B-Instruct | ✗ |
| Llama-3.2-1B | Llama-3.2-1B-Instruct | ✗ |
| Llama-3.2-3B | Llama-3.2-3B-Instruct | ✗ |
| Qwen3-0.6B-Base | Qwen3-0.6B | ✓ |
| Qwen3-8B-Base | Qwen3-8B | ✓ |
| Qwen3-14B-Base | Qwen3-14B | ✓ |
| Qwen3-30B-A3B-Base | Qwen3-30B-A3B | ✓ |

Recent work finds that such a post-training process is superficial (Zhou et al., 2023a; Ye et al., 2025; Ji et al., 2025; Wu et al., 2025a). Superficial Alignment Hypothesis claims that the model's knowledge and capabilities are acquired almost entirely during pretraining, while alignment teaches it which sub-distribution of formats should be used when interacting with users (Zhou et al., 2023a; Ye et al., 2025). Model Elasticity finds that models tend to maintain the original distribution, i.e., resist alignment and return quickly when tuned in the opposite direction (Ji et al., 2025). Shadow-FT represents that paired Base and Instruct are highly similar in weights (Wu et al., 2025a). Compared to these previous works, we revisit Base and Instruct models from the view of effective rank, providing a novel perspective on the superficiality.

**Effective Rank.** Effective rank (eRank) measures the uniformity of the singular value distribution to quantify the effective dimensionality of a matrix (Roy & Vetterli, 2007). For any non-zero matrix $\mathbf{W} \in \mathbb{R}^{d_1 \times d_2}$ with singular values $\Sigma = \{\sigma_1, \sigma_2, \sigma_3, ..., \sigma_{r-1}, \sigma_r\}$ where $r = \min\{d_1, d_2\}$. The eRank of $\mathbf{W}$ is defined as the exponential of the Shannon entropy computed from its normalized singular value distribution, formulated as follows:

$$\text{eRank}(\mathbf{W}) = \exp\left(-\sum_{i=1}^{r} \frac{\sigma_i^{\gamma}}{\sum_{i=1}^{r} \sigma_i^{\gamma}} \log\left(\frac{\sigma_i^{\gamma}}{\sum_{i=1}^{r} \sigma_i^{\gamma}}\right)\right). \tag{1}$$

The $\gamma$ is the scale factor and can be 1 or 2, typically. In this paper, we set $\gamma = 1$. Meanwhile, we can calculate the entropy using $\log_2(x)$ and then apply $2^x$ instead of $\exp(x)$ in Equation 1. We can easily prove that the eRank would be exactly the same.

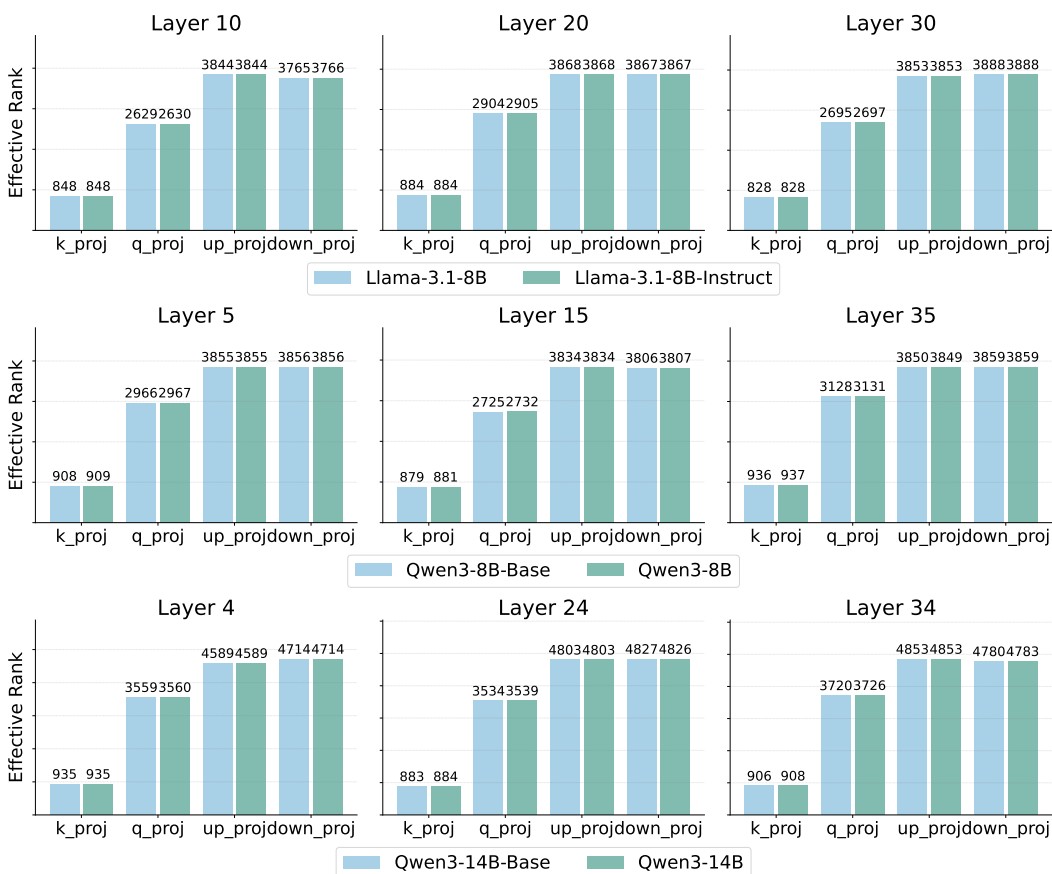

Figure 1: Effective ranks (eRank) of the linear weights from various paired Base and Instruct models. We randomly select three layers, and `k_proj` is relatively small due to the Grouped-Query Attention (GQA) mechanism. We can find that eRanks from paired Base and Instruct models are almost the same.

Effective rank plays an important role in measuring the information among hidden states. Diff-eRank assesses LLMs by analyzing hidden representations and measuring how efficiently LLMs eliminate redundant information during training (Wei et al., 2024). Li et al. (2025) employs the eRank of gradients to assess the quality of training data. To the best of our knowledge, we are the first to directly analyze the eRank of weights.

## 2.2 Revisit Base and Instruct via Effective Rank

To investigate the effects of post-training, we examine the weights of Base and Instruct models through the lens of effective rank (eRank). Our analysis covers several mainstream LLMs, including the Llama and Qwen3 series. Without loss of generality, we randomly select three representative linear layers from the bottom, middle, and top. Due to space constraints, we report the ceiling of the eRank values.

Figure 1 shows the paired eRank distributions. The core finding is that the eRank values for corresponding linear layers in the Base and Instruct models are nearly identical. For instance, the k_proj matrix in Layer 4 of Qwen3-14B has an eRank of 935 in both Base and Instruct versions. This striking similarity holds across all models and layers tested, demonstrating the robustness of this phenomenon.

Given that eRank quantifies the effective dimensionality of a weight matrix, our results indicate that this dimensionality remains almost unchanged after post-training. Therefore, we can conclude that

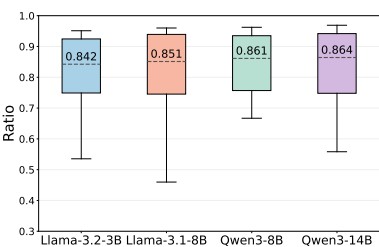 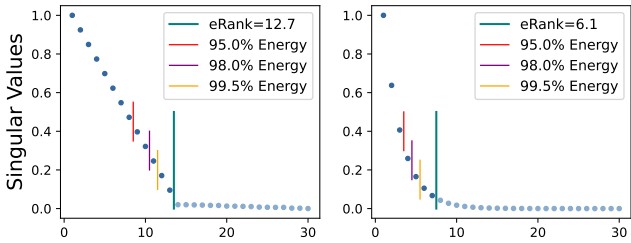

Figure 2: The distribution of eRank-to-Rank ratios for all linear layers in various LLMs.

Figure 3: Toy examples of singular value distributions and thresholds using eRank or energy. eRank works well as an *adaptive* threshold.

the knowledge acquired during pre-training is retained, reinforcing the hypothesis that post-training is a superficial process.

We further analyze the distribution of the eRank-to-Rank ratio, defined as $\mathrm{eRank}/r$. By definition (Equation 1, $1 \leq \mathrm{eRank} \leq r$), this ratio is constrained to the interval (0,1]. As illustrated in Figure 2, the ratios are highly concentrated. For all evaluated models, the mean ratio remains stable at approximately 0.85, and the interquartile range consistently falls between 0.75 and 0.95. This suggests that eRank is consistently a high fraction of the total rank.

## 3 METHODOLOGY

### 3.1 MOTIVATION

**Challenge.** The post-training phase also introduces a critical trade-off between exploitation and exploration. During this stage, the Instruct model is optimized to maximize rewards by focusing on the most effective reasoning paths, thereby sharpening its exploitative capabilities. However, this intense focus comes at the cost of its ability to explore a diverse range of solutions. This trade-off is empirically evident in model performance. Specifically, Instruct models significantly outperform their base counterparts on Pass@1, but tend to underperform on Pass@k for larger values of k (Yue et al., 2025; Wang et al., 2025a; Zhu et al., 2025). Therefore, how to enhance the exploration without compromising its exploitation remains a challenge.

**Refine Instruct with Base.** Recent works have sought to mitigate this limitation by introducing additional training objectives (Chen et al., 2025) or external tools (Wang et al., 2025a). These methods, however, invariably incur significant computational overhead during training or inference. In this paper, we focus on the training-free method to circumvent this issue.

Given that the post-training process is superficial at the weight level (Wu et al., 2025a), one intuitive idea is to enhance Instruct with the weights from the Base model. Such a training-free strategy has been validated on related tasks such as model merge (Yang et al., 2024; Zhang et al., 2024; Wu et al., 2025b). The Base model contains almost all the knowledge, while the Instruct model only elicits part of the high-reward thinking patterns. As the post-training shrinks the generative horizon (Yang & Holtzman, 2025), our solution is to partially revert the Instruct model towards its Base state. Therefore, our next goal is to refine the weight deltas between the Instruct and Base models.

### 3.2 TIMBER

To refine the weight deltas, one naive way is to scale them linearly. However, the modifications from post-training are known to be fragile (Ji et al., 2025), and this simple scaling often fails (see Section 5.2). Fortunately, eRank measures the effective dimension of the matrix and suitable for indicating the majority of the singular values. For instance, eRank would be 1 for singular values $\Sigma = \{1, 0, ..., 0, 0\}$ and be $r$ for $\Sigma = \{1, 1, ..., 1, 1\}$. As illustrated in the toy examples in Figure 3, eRank serves as an effective threshold for isolating the principal components of the singular value spectrum.

Motivated by this property, we propose a simple yet effective training-free method named **Timber**. The core idea is to enhance the weight delta via partially reverting the Instruct model towards its Base state. Specifically, we employ eRank as a threshold to partition the singular values (i.e., the matrix spectrum) of the weight delta into head and tail parts, and then either remove or attenuate the tail.

For the weight matrices $\mathbf{W}_B \in \mathbb{R}^{m \times n}$ and $\mathbf{W}_I \in \mathbb{R}^{m \times n}$ from the same linear layer of Base and Instruct models, we first compute the weight delta:

$$\mathbf{W_\Delta} = \mathbf{W}_I - \mathbf{W}_B. \tag{2}$$

The weight delta $\mathbf{W_\Delta}$ is typically full-rank and rank $r = \min\{m, n\}$ We then calculate the singular values via SVD decomposition:

$$\mathrm{SVD}(\mathbf{W_\Delta}) \rightarrow \mathbf{U}\mathbf{\Sigma}\mathbf{V}^T, \tag{3}$$

where $\mathbf{U}$ and $\mathbf{V^T}$ are two orthogonal matrices and $\mathbf{\Sigma} = \mathrm{diag}(\sigma_0, \sigma_1, ..., \sigma_{r-1}, \sigma_r)$ contains the singular values in non-decreasing order. Our goal is to create a refined weight matrix $\mathbf{W}_I^+$ by modifying these singular values:

$$\mathbf{W}_I^+ = \mathbf{W}_B + \mathrm{refine}(\mathbf{W_\Delta}) = \mathbf{W}_B + \mathbf{U} \, \mathrm{refine}(\mathbf{\Sigma})\mathbf{V}^T, \tag{4}$$

where $\mathrm{refine}(\cdot)$ is the enhancement operation to the singular values in $\mathbf{\Sigma}$.

In Timber, we define this refinement process as follows. First, we set a threshold K based on the ceiling of the eRank:

$$K := \lceil \mathrm{eRank}(\mathbf{W_\Delta}) \rceil. \tag{5}$$

One strategy is to remove the tail part by zeroing out singular values beyond the $K-$th position:

$$\mathrm{refine}(\mathbf{\Sigma}) = \mathrm{diag}\{\underbrace{\sigma_1, \sigma_2, ..., \sigma_K}_{\text{Top-K,preserve}}, \underbrace{0, ..., 0, 0}_{discard}\}. \tag{6}$$

Since this operation lowers the rank of the weight delta, we name this strategy **Timber-L**. As shown in Figure 2, the eRanks are around the 85th percentile of full rank.

Another strategy is to attenuate the tail part rather than discard it entirely:

$$\mathrm{refine}(\mathbf{\Sigma}) = \mathrm{diag}\{\underbrace{\sigma_1, \sigma_2, ..., \sigma_K}_{\text{Top-K,preserve}}, \underbrace{\lambda \cdot \sigma_{K+1}, ..., \lambda \cdot \sigma_{r-1}, \lambda \cdot \sigma_r}_{attenuate}\}, \tag{7}$$

where $0 < \lambda < 1$ is an attenuation factor. This full-rank strategy is referred to as Timber. Note that Timber-L is a special case of this approach where $\lambda = 0$. When applying Timber, we only modify the weights of linear layers, leaving bias terms and normalization layers unchanged.

## 4 EXPERIMENTS

### 4.1 EXPERIMENTAL SETUP

**Models.** We conduct experiments on mainstream LLMs, specifically the Llama 3 and Qwen3 series, with model sizes ranging from 0.6B to 30B. In particular, we also include the MoE-style Qwen3-30B-A3B for a more comprehensive setting. All model checkpoints are downloaded from the official HuggingFace repository. Detailed model information is provided in Table 1. For Timber, we search for the best attenuation factor in $\{0.2, 0.5, 0.8\}$ based on the performance on AIME'24. This search incurs a minimal computational cost as Timber is a **training-free** method. Please refer to Appendix A.4 for the detailed score with different $\lambda$.

**Evaluation.** We evaluate the models on a suite of mainstream benchmarks spanning various tasks: IFEval (Zhou et al., 2023b) for the instruction following, MATH (Hendrycks et al., 2021) and MATH-500 (Lightman et al., 2023) for mathematical reasoning, GPQA-Diamond (GPQA-D, (Rein et al., 2024)) for scientific question answering, and HellaSwag (Zellers et al., 2019) for common-sense reasoning. To assess the Qwen3 series in Thinking mode, we also utilize the challenging AIME'24 [1] and HumanEval (Chen et al., 2021) for coding task. For all models, we use the officially recommended hyperparameters for inference. Further details on the benchmarks and evaluation settings can be found in Appendix A.3.

---

[1]https://huggingface.co/datasets/AI-MO/aimo-validation-aime

Table 2: Performance of vanilla Instruct, proposed Timber-L, and Timber on mainstreaming benchmarks regarding Llama and Qwen3 series. The Qwen3 models are evaluated in *Non-thinking* mode. Under all the settings, Timber-L and Timber outperform the baseline **without any training**.

| Model | Setting | IFEval | MATH-500 | MATH | GPQA-D | HellaSwag | Avg. | Δ |
|---|---|---|---|---|---|---|---|---|
| Llama-3.2-1B | Instruct | 48.34 | 14.85 | 16.88 | 22.98 | 34.94 | 27.60 | |
| | Timber-L | 49.17 | 15.20 | 16.77 | 25.88 | 35.77 | **28.56** | +0.96 |
| | Timber | 49.58 | 14.95 | 17.08 | 24.62 | 35.35 | 28.32 | +0.72 |
| Llama-3.2-3B | Instruct | 69.64 | 41.35 | 34.22 | 24.75 | 61.01 | 46.19 | |
| | Timber-L | 68.81 | 41.70 | 34.48 | 25.76 | 61.29 | 46.41 | +0.22 |
| | Timber | 69.59 | 42.05 | 34.23 | 25.51 | 61.46 | **46.57** | +0.38 |
| Llama-3.1-8B | Instruct | 74.25 | 49.60 | 39.46 | 28.62 | 75.65 | 53.52 | |
| | Timber-L | 73.81 | 51.13 | 39.21 | 28.96 | 81.36 | 54.89 | +1.37 |
| | Timber | 75.66 | 50.67 | 39.35 | 28.11 | 82.37 | **55.23** | +1.71 |
| Qwen3-0.6B | Instruct | 56.75 | 52.00 | 41.92 | 23.74 | 43.42 | 43.57 | |
| | Timber-L | 56.75 | 52.60 | 42.66 | 23.40 | 43.27 | 43.74 | +0.17 |
| | Timber | 57.36 | 51.60 | 43.11 | 28.79 | 43.30 | **44.83** | +1.26 |
| Qwen3-8B | Instruct | 82.75 | 84.20 | 73.27 | 45.62 | 84.62 | 74.09 | |
| | Timber-L | 83.46 | 84.00 | 73.21 | 46.63 | 84.67 | 74.39 | +0.30 |
| | Timber | 83.09 | 84.73 | 73.23 | 48.48 | 84.68 | **74.84** | +0.75 |
| Qwen3-14B | Instruct | 85.15 | 87.47 | 75.74 | 47.81 | 88.13 | 76.86 | |
| | Timber-L | 85.77 | 86.60 | 75.97 | 51.68 | 87.97 | 77.60 | +0.74 |
| | Timber | 85.46 | 88.00 | 75.77 | 51.01 | 88.04 | **77.66** | +0.80 |
| Qwen3-30B-A3B | Instruct | 84.17 | 88.00 | 76.99 | 44.95 | 89.25 | 76.67 | |
| | Timber-L | 85.71 | 88.33 | 77.00 | 45.12 | 89.43 | 77.12 | +0.45 |
| | Timber | 85.09 | 88.80 | 76.85 | 45.96 | 89.27 | **77.19** | +0.52 |

**Metric.** To evaluate the exploration capability, we use the popular Pass@k, which is defined as the fraction of problems for which at least one correct response is produced in $k$ independent trials. However, directly computing Pass@k using only $k$ rollouts per problem often suffers from high variance. Therefore, we employ the unbiased estimator (Chen et al., 2021). Specifically, we roll out for $n$ times ($n \geq k$), and calculate Pass@k as follows:

$$\text{Pass@}k := \mathbb{E}_{x \sim \mathcal{D}} \left[ 1 - \frac{\binom{n-c}{k}}{\binom{n}{k}} \right], \tag{8}$$

where $x$ is the input prompt from dataset $D$, and $c$ is the count of correct solutions. In addition to Pass@k, we also report Mean@k, defined as the average accuracy across $k$ independent trials. We repeat 4 times for Llama-3.2-1B, and 3 times for the rest larger models.

## 4.2 MAIN RESULTS

As shown in Table 2, we report the Mean@k results on 6 benchmarks and their average score. We repeat 4 trials for Llama-3.2-1B (k=4) and 3 for others (k=3). Some findings can be concluded as follows:

- *Our proposed Timber consistently and comprehensively outperforms the vanilla Instruct model.* Across all tested models, both Timber-L and Timber significantly outperformthe vanilla Instruct model. For instance, Timber achieves an average score of 55.23 for Llama-3.1-8B, which is 1.71 higher than baseline.

- *The attenuation strategy of Timber is generally superior to directly dropping in Timber-L.* When comparing the two proposed variants, the standard Timber method demonstrates a greater performance gain (i.e., Δ) than Timber-L in 6 out of 7 cases. For instance, on Qwen3-0.6B, Timber gets an average of 44.83, significantly higher than 43.74 of Timber-L. This suggests that Timber strikes a better balance between optimization and knowledge preservation.

- *Timber is a robust and broadly applicable training-free plug-in*. The performance benefits of Timber are not limited to a specific model family, size, or architecture. The method

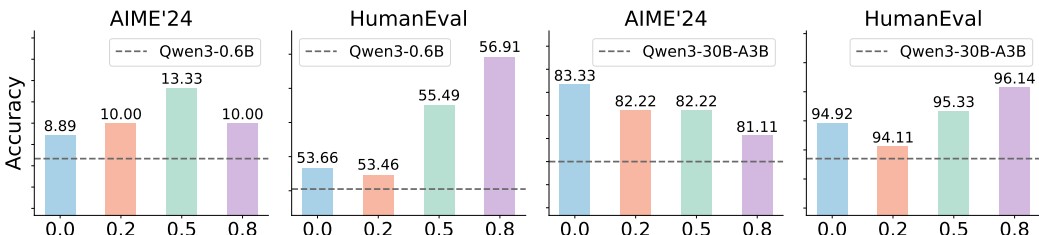

Figure 4: Accuracy on AIME'24 and HumanEval benchmarks for Timber with various $\lambda$. The vanilla scores for Instruct models are 6.67, 53.05, 80.0, and 93.7, respectively. For both models, we sample the results under *Thinking* mode. Timber shows strong robustness regarding $\lambda$.

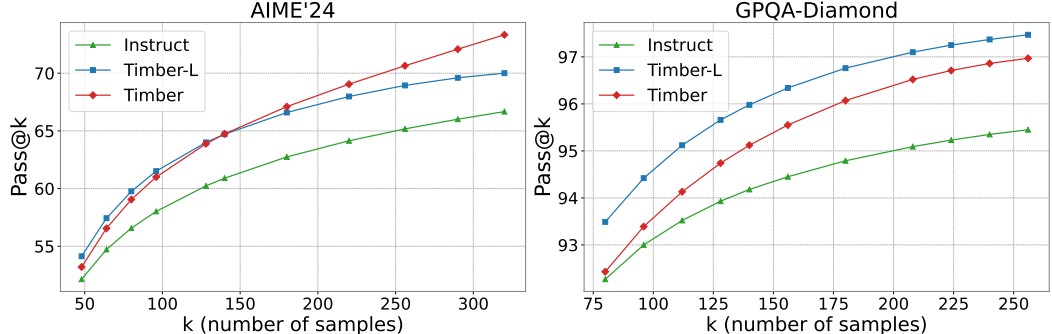

Figure 5: Pass@k results on AIME'24 and GPQA-Diamond benchmarks for Qwen3-0.6B under Thinking mode. Both Timber and Timber-L improve the exploration significantly. Please refer to Appendix A.4 for more detailed results.

proves effective on both the Llama 3 and Qwen3 series and scales from small 0.6B models to the large 30B Mixture-of-Experts (MoE) architecture.

The Qwen3 series supports the hybrid thinking, allowing them to generate outputs in either Thinking or Non-Thinking mode. In the Thinking mode, the LLM will output a longer reasoning process and typically performs better thanks to the test-time scaling. To assess their advanced reasoning capabilities, we further evaluate the thinking capability of Qwen3-0.6B and Qwen3-30B-A3B on the AIME'24 and HumanEval tasks. As shown in Figure 4, our proposed method, Timber, consistently outperforms the vanilla Instruct model across various attenuation factors $\lambda$, demonstrating both superior effectiveness and robustness. For instance, when applied to Qwen3-30B-A3B, Timber achieves a score of 96.14 on HumanEval, surpassing the vanilla Instruct model by 2.44 points.

The core principle of Timber is to partially revert the Instruct model towards its Base state, a process designed to enhance exploration without compromising exploitation. To validate this hypothesis, we conducted further experiments on Qwen3-0.6B, evaluating its exploration performance using the Pass@k metric. During inference, we configured the model to rollout in Thinking mode, generating 320 candidate samples for AIME'24 and 256 for GPQA-Diamond. The results, shown in Figure 5, indicate that both Timber and Timber-L achieve significantly higher Pass@k scores than the vanilla Instruct model.

Critically, the performance gap between the Timber methods and the Instruct baseline widens as $k$ increases. This trend provides strong evidence that Timber is fundamentally more effective at exploring the solution space and generating a diverse set of high-quality candidates. In summary, our proposed *training-free* method, Timber, successfully enhances exploration without compromising exploitation, a conclusion supported by the comprehensive results in Table 2 and Figure 4.

Table 3: Performance for Truncated SVD methods and proposed Timber on Qwen3-8B. Truncate-R denotes the strategy on the ratio of full rank, and Truncated-E on the ratio of energy.

| Method | Energey | IFEval | MATH-500 | MATH | GPQA-D | HellaSwag | Avg. |
|---|---|---|---|---|---|---|---|
| Instruct | 100.00% | 82.75 | 84.20 | 73.27 | 45.62 | 84.62 | 74.09 |
| Truncate-R | 99.60% | 83.30 | 85.07 | 73.21 | 44.95 | 84.74 | 74.25 |
| | 99.04% | 83.18 | 83.87 | 73.08 | 47.47 | 84.69 | 74.46 |
| | 98.30% | 82.50 | 85.00 | 72.99 | 44.11 | 84.62 | 73.84 |
| | 97.39% | 83.43 | 84.60 | 73.56 | 46.63 | 84.70 | 74.58 |
| Truncate-E | 99.50% | 83.30 | 84.27 | 72.99 | 44.28 | 84.73 | 73.91 |
| | 99.00% | 82.62 | 84.33 | 72.73 | 45.96 | 84.72 | 74.07 |
| | 98.00% | 84.47 | 84.47 | 72.93 | 44.95 | 84.76 | 74.32 |
| | 95.00% | 83.49 | 83.87 | 73.23 | 44.95 | 84.87 | 74.08 |
| Timber (Ours) | 98.82% | 83.09 | 84.73 | 73.23 | 48.48 | 84.68 | **74.84** |

## 5 EXTENSIVE ANALYSIS

### 5.1 COMPARISON WITH TRUNCATED SVD

Truncated SVD is a widely employed technique for compressing Large Language Models (LLMs) (Wang et al., 2025b; Qinsi et al., 2025). While our method is conceptually different, Timber-L can be interpreted as a special case of SVD applied to the weight deltas, where truncation occurs at the effective rank (eRank). However, Timber differs fundamentally: instead of discarding the tail singular values, it attenuates them with a scaling factor.

For further comparison, despite these theoretical differences, we design two SVD truncation baselines: Truncate-R and Truncate-E. Truncate-R discards singular values based on a fixed ratio of the full rank, while Truncate-E based on a target energy preservation ratio. The energy of the singular value distribution is a vital metric representing the amount of preserved information. Therefore, we set the ratios in Truncate-R and Truncate-E to be comparable with our proposed method, Timber. Specifically, Timber preserves 98.82% of the total energy on Qwen3-8B. For Truncate-R, we thus set the rank ratios to 0.95, 0.9, 0.85, and 0.8, which correspond to preserving 99.60%, 99.04%, 98.30%, and 97.39% of the total energy, respectively. For Truncate-E, we set the energy preservation thresholds directly to 99.50%, 99.00%, 98.00%, and 95.00%.

Table 3 details the performance of these methods on benchmarks. We can find that our proposed Timber method achieves the highest average score (74.84), outperforming all variants of the Truncated SVD baselines. While both Truncate-R and Truncate-E can outperform the Instruct model, the performance is unstable and highly sensitive to the truncation threshold. In contrast, Timber provides a more substantial and robust performance gain, suggesting that eRank serves as a solid threshold and attenuating the tail singular values is more effective than discarding.

### 5.2 COMPARISON WITH MODEL MERGE

Model merging is a training-free paradigm that combines the weights of specialized models to create a single, more capable one (Yang et al., 2024). Our method, Timber, utilizes a similar principle to refine the weights of Instruct model by incorporating weights from paired Base model. To benchmark our approach, we compare Timber against a straightforward model merging strategy: direct weighted averaging of the model parameters. Specifically, we merge the weights of layers using a linear interpolation:

$$\mathbf{W_{merge}} = \mu\mathbf{W}_I + (1 - \mu)\mathbf{W}_B = \mathbf{W}_B + \mu\mathbf{W}_\Delta, \qquad (9)$$

where $\mathbf{W}_\Delta$ denotes the weight difference and $\mu$ is a global scaling factor. From this perspective, simple model merging is a special case that applies a uniform linear scale to the entire weight delta. In contrast, Timber employs a more sophisticated, fine-grained paradigm that scales the weight delta based on its eRank.

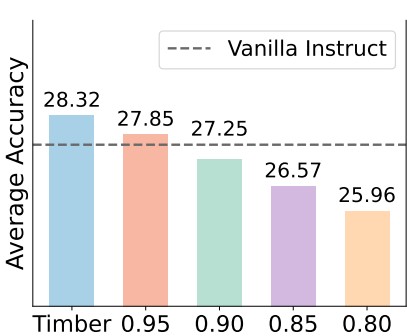 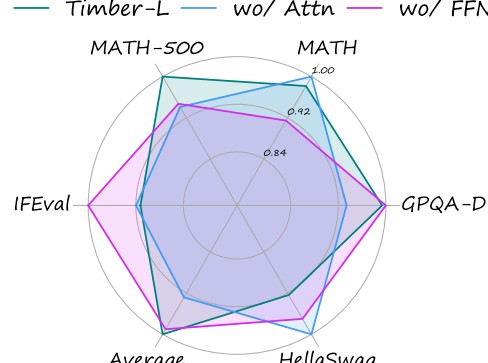

Figure 6: Performance of Timber and model merge strategies with various $\mu$ on Llama-3.2-1B. Vanilla Instruct gets a score of 27.60.

Figure 7: Normalized scores of Timber and its ablation on the module to apply. *Attn* and *FFN* denote the attention and FFN layers.

Figure 6 shows the results on Llama-3-1B. The simple merging strategy can slightly improve performance over the Vanilla Instruct baseline when $\mu$ is 0.95. As the scaling factor $\mu$ decreases further, performance degrades sharply, quickly falling below the baseline. This highlights the fragility of applying a uniform scaling factor. In contrast, Timber, a fine-grained refinement based on eRank, shows a more robust and significant performance enhancement, demonstrating the superiority of its more nuanced merging strategy.

### 5.3 MODULES TO APPLY TIMBER

Prior research indicates that in Transformer, FFN layers primarily store factual and commonsense knowledge, while attention layers are responsible for mixing information between tokens within the context (Geva et al., 2021; Dai et al., 2022; Meng et al., 2023). Given these distinct roles, we conducted an ablation study to isolate which module benefits most from the Timber method.

Figure 7 presents the results of applying Timber-L without attention layers (wo/ Attn) or without FFN layers (wo/ FFN) on Llama-3.1-8B. The scores are normalized for better visualization. We can find that applying Timber-L to both modules yields the best overall performance, particularly on knowledge-intensive tasks like MATH and GPQA-D. Meanwhile, reverting the attention module only (wo/ FFN) performs better at IFEval, while reverting the FFN module only (wo/ Attn) benefits the math reasoning tasks. This suggests that the complex logical flows required for reasoning rely heavily on the attention weights, whereas the knowledge-intensive capabilities are predominantly stored in the FFN layers. Such observation is consistent with the conclusion that FFN modules primarily store factual knowledge (Geva et al., 2021; Dai et al., 2022), while attention modules are responsible for information mixing.

### 5.4 CASE STUDY

We further analyze the cases of generated responses. Please refer to Appendix A.6 for detailed examples and analysis on Qwen3-14B. Timber outperforms Instruct model with more comprehensive thinking trajectories. In short, Timber can effectively refine Instruct with Base and thus achieve a better trade-off between exploration and exploitation, which is consistent with the conclusions in Section 4.2.

## 6 CONCLUSION

In this work, we first carefully compare the Base and Instruct models in terms of effective rank, reinforcing the hypothesis that post-training is superficial. To tackle the issue that the exploration capability of the Instruct model is limited, we further propose a simple yet effective training-free method, Timber, to refine the weight delta. The key is to partially revert the Instruct model towards its Base state. Specifically, we first employ eRank as a threshold to split the singular values of weight

deltas, followed by an enhancement strategy that either removes or attenuates the tail. Extensive experiments show that Timber successfully enhances exploration without compromising exploitation. We leave it for future work to explore more strategies to enhance the weight deltas. Also, applying Timber to other model merge scenarios is another interesting topic.

## REPRODUCIBILITY STATEMENT

We are committed to the reproducibility of our work. The full source code required to reproduce our main findings is included in the *supplementary material*. Corresponding hyperparameters and detailed configuration files for all experiments are documented in Appendix A.3.2. All experiments were conducted on publicly available benchmarks, and the details are provided in Appendix A.3. All the models will be made public in the future.

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

# A APPENDIX

## A.1 LLM USAGE

We utilized a large language model (LLM) to assist in proofreading and refining the language of our manuscript. The use was limited to improving clarity, grammar, and style for both the main text and figure captions. We authors are fully responsible for all scientific claims and the final content of this paper.

## A.2 MORE RELATED WORK

**Effective Rank.** The effective rank (eRank) is a metric used to quantify the flatness of a singular value distribution, offering insights into the intrinsic dimensionality of a representation. It has found diverse applications in analyzing large language models (LLMs). For instance, Diff-eRank utilizes the eRank of hidden representations to measure how efficiently LLMs prune redundant information during training (Wei et al., 2024). Beyond analysis, eRank has been adapted to measure data quality by examining gradients (Li et al., 2025) and to guide network design, as seen in the KRAdapter fine-tuning method (Albert et al., 2025) and the Stiefel optimizer (Park et al., 2025). In our work, we employ eRank to provide further evidence on the nature of post-training adjustments in LLMs.

**Weight-Level Similarity of Base and Instruct Models.** Instruct models are derived from Base models via post-training and thus share an identical architecture. Research has shown that their similarity extends to the weight level, where the differences are often minimal. For example, RL updates only a small subnetwork, leaving most parameters unchanged (Mukherjee et al., 2025). Corroborating this, Wu et al. (2025a) demonstrated that the weight difference between a Base model and its paired Instruct model can be less than 5%. Furthermore, Base and Instruct models have been observed to exhibit similar emergent behaviors, such as similar training dynamics during RL training (Xie et al., 2025) and similar attention sink phenomena (Gu et al., 2025). Building on these findings, our paper, to the best of our knowledge, is the first to analyze the similarity between Base and Instruct models from the perspective of effective rank.

## A.3 EVALUATION DETAILS

### A.3.1 BENCHMARK

We conduct evaluation on the wonderful framework OpenCompass (Contributors, 2023). More details about the evaluated benchmark are as follows:

- IFEval (Zhou et al., 2023b): evaluating instruction-following language models, focusing on their ability to understand and respond to various prompts. It includes 25 types of those verifiable instructions and is constructed around 500 prompts, with each prompt containing one or more verifiable instructions. We report the prompt_level_strict accuracy under a zero-shot setting.

- MATH (Hendrycks et al., 2021): evaluating the mathematical reasoning abilities of AI models through a variety of problem types, including arithmetic, algebra, geometry, and more. There are 7,500 training examples and 5000 test samples. We report the accuracy under a 4-shot setting.

- MATH-500 (Lightman et al., 2023): 500 uniformly selected test problems from MATH. We report the accuracy under a 4-shot setting.

- GPQA-Diamond (GPQA-D, (Rein et al., 2024)): evaluating the reasoning ability of large language models (LLMs) on challenging multiple-choice questions written by domain experts in biology, physics, and chemistry. It contains 198 selected questions that require step-by-step reasoning to arrive at the correct answer. We report the accuracy under the 0-shot setting.

- HellaSwag (Zellers et al., 2019): evaluating the ability on commonsense reasoning tasks. It consists of multiple-choice questions where the model must select the most plausible continuation of a given context. We report the accuracy under the 0-shot setting.

- AIME'24 [2]: evaluating the ability to solve challenging mathematics problems from the American Invitational Mathematics Examination, a prestigious high school mathematics competition. We report the accuracy under the 0-shot setting.

- HumanEval (Chen et al., 2021): evaluating the ability of code generation models to write Python functions based on given specifications. It includes 164 programming problems with a function signature, docstring, body, and several unit tests.

### A.3.2 HYPERPARAMETER FOR GENERATION

We follow the official recommended hyperparameters for inference. The details as shown in Table 4. We turn on the sampling strategy for more diversity.

Table 4: Hyperparameter during generation for different models.

| Model | Temperature | Top_p | Top_k | Max_token |
|---|---|---|---|---|
| Llama-3.2-1B | 0.6 | 0.9 | - | 4096 |
| Llama-3.2-3B | 0.6 | 0.9 | - | 8192 |
| Llama-3.1-8B | 0.6 | 0.9 | - | 8192 |
| Qwen3 Series (Non-Thinking) | 0.7 | 0.8 | 20 | 8192 |
| Qwen3 Series (Thinking) | 0.6 | 0.95 | 20 | 38912 |

### A.4 PERFORMANCE WITH DIFFERENT $\lambda$

Table 7 shows the results of Timber with different $\lambda$. We can find that Timber consistently outperforms the vanilla baseline, demonstrating the robustness. Also, $\lambda = 0.2$ is a sweet point. We recommend setting $\lambda$ to 0.2 for the latest released models.

### A.5 DETAILED RESULTS UNDER THINKING MODE

Table 5: Detailed Pass@k performance on AIME24.

| Model | Pass@1 | Pass@4 | Pass@8 | Pass@32 | Pass@64 | Pass@128 | Pass@256 | Pass@320 |
|---|---|---|---|---|---|---|---|---|
| Instruct | 10.79% | 23.14% | 31.20% | 48.08% | 54.72% | 60.23% | 65.17% | 66.67% |
| Timber-L | 11.04% | 23.12% | 30.75% | **49.02%** | **57.43%** | **64.00%** | 68.94% | 70.00% |
| Timber | **11.39%** | **23.78%** | **31.50%** | 48.36% | 56.55% | 63.88% | **70.64%** | **73.33%** |

Table 6: Detailed Pass@k performance on GPQA-Diamond.

| Model | Pass@1 | Pass@4 | Pass@8 | Pass@16 | Pass@32 | Pass@64 | Pass@128 | Pass@256 |
|---|---|---|---|---|---|---|---|---|
| Instruct | 26.02% | 53.60% | 67.21% | 78.28% | 86.25% | 91.21% | 93.93% | 95.45% |
| Timber-L | 25.78% | 54.15% | 67.83% | 78.71% | **86.73%** | **92.18%** | **95.66%** | **97.47%** |
| Timber | **26.04%** | **54.38%** | **68.03%** | **78.72%** | 86.17% | 91.15% | 94.74% | 96.97% |

As shown in Table 5 and 6, for almost all the k values, Timber/Timber-L outperforms the Instruct model from Pass@1 to Pass@320/256. For AIME24 (Pass@1), Timber improves from 10.79% (Instruct) to 11.39%. For GPQA (Pass@1), Timber maintains performance (26.04% vs 26.02%). The performance gap widens significantly as $k$ increases. In conclusion, Timber-L and Timber demonstrate consistent performance improvements, particularly in preserving exploitation at lower $k$ values while enhancing exploration at higher $k$ values.

---

[2]https://huggingface.co/datasets/AI-MO/aimo-validation-aime

A.6   DETAILED CASES

We showcase specific questions from GPQA-Diamond and corresponding answers from Qwen3-14B. In particular, we employ the Gemini 2.5 pro to simplify the answers for better visualization. The prompt for generation is:

> Answer the following multiple-choice question. The last line of your response should be of the following format: 'ANSWER: LETTER' (without quotes), where LETTER is one of ABCD. Think step by step before answering.

The proposed Timber outperforms vanilla Instruct model regarding the more comprehensive thinking trajectory (shown in Table 8, Table 10) and more extensive domain knowledge (shown in Table 9).

Table 7: The average of 5 benchmarks for vanilla Instruct and Timber with different $\lambda$.

| Model | Vanilla | Timber | | | |
|---|---|---|---|---|---|
| | | $\lambda = 0.0$ | $\lambda = 0.2$ | $\lambda = 0.5$ | $\lambda = 0.8$ |
| Llama-3.2-1B | 27.60 | 28.56 | 28.32 | 27.91 | 27.69 |
| Llama-3.2-3B | 46.19 | 46.41 | 46.23 | 46.57 | 46.49 |
| Llama-3.1-8B | 53.52 | 54.89 | 55.23 | 53.84 | 53.16 |
| Qwen3-0.6B | 43.57 | 43.74 | 43.86 | 43.92 | 44.83 |
| Qwen3-8B | 74.09 | 74.39 | 74.84 | 74.46 | 74.12 |
| Qwen3-14B | 76.86 | 77.60 | 77.66 | 77.31 | 77.43 |
| Qwen3-30B-A3B | 76.67 | 77.12 | 76.82 | 77.19 | 77.19 |

Table 8: One case from GPQA-Diamond benchmark. The responses from Timber-L and Timber are more comprehensive via traversal each answer.

| Question | You have prepared a di-substituted 6-membered aromatic ring compound. The FTIR spectrum of this compound shows absorption peaks indicating the presence of an ester group. The 1H NMR spectrum shows six signals: two signals corresponding to aromatic-H, two signals corresponding to vinyl-H (one doublet and one doublet of quartets), and two signals corresponding to –CH3 groups. There are no signals corresponding to –CH2 groups. Identify the chemical formula of this unknown compound as either C11H12O2, C11H14O2, C12H12O2, or C12H14O2. A) C11H14O2 B) C12H12O2 C) C12H14O2 D) C11H12O2 |
|---|---|
| Answer | D |
| Instruct | Based on the spectroscopic data, the compound is a di-substituted aromatic ring containing an ester group, a vinyl group, and two methyl groups. The atom count is determined as follows: Carbon: 6 (aromatic ring) + 2 (vinyl) + 2 (two methyls) + 2 (ester) = 12 C Hydrogen: 4 (aromatic) + 2 (vinyl) + 6 (two methyls) = 12 H Oxygen: 2 (ester) = 2 O This leads to the chemical formula C12H12O2. **ANSWER: B** |
| Timber-L | Based on the spectral data, the compound has a di-substituted aromatic ring, an ester group, a vinyl group, and two methyl groups, with no methylene (-CH2) groups. A plausible structure fitting these clues is a benzene ring substituted with a methyl ester group (-COOCH3) and a propenyl group (-CH=CHCH3). Let's count the atoms for this structure: Carbon (C): 6 (from the ring) + 1 (from C=O) + 1 (from -OCH3) + 3 (from the propenyl group) = 11 C Hydrogen (H): 4 (from the ring) + 3 (from -OCH3) + 5 (from the propenyl group) = 12 H Oxygen (O): 2 (from the ester group) = 2 O The resulting chemical formula is C11H12O2. which matches option D. **ANSWER: D** |
| Timber | Based on the structural information, the degree of unsaturation (DoU) is calculated first. An aromatic ring contributes 4 DoU. An ester group (C=O) contributes 1 DoU. A vinyl group (C=C) contributes 1 DoU. Total required DoU = 4 + 1 + 1 = 6. Next, we calculate the DoU for each option: A) C11H14O2: DoU = 5 B) C12H12O2: DoU = 7 C) C12H14O2: DoU = 6 D) C11H12O2: DoU = 6 Only options C and D have the correct DoU of 6. To decide between them, we analyze the $^1$H NMR data. A di-substituted benzene ring has 4 H, the vinyl group has 2 H, and two methyl groups have 6 H (2 × 3 H). Total H = 4 (aromatic) + 2 (vinyl) + 6 (methyls) = 12 H. Comparing this to the remaining options, only C11H12O2 has 12 hydrogens. A possible isomer that fits this data is methyl 4-(prop-1-en-1-yl)benzoate. **ANSWER: D** |

Table 9: One case from GPQA-Diamond benchmark. The vanilla Instruct model lacks the relevant domain knowledge

| | |
|---|---|
| Question | ChIP-seq on a PFA-fixed sample with an antibody to the IKAROS transcription factor in human B cells followed by next-generation sequencing and standard quality control, alignment and peak-calling steps produced ChIP peaks that disappeared when PFA+DSG fixation was used. Where are we most likely to find such disappearing peaks?A) At repeats B) In the introns of large genes C) At active promoters and enhancers D) At random locations in the genome |
| Answer | C |
| Instruct | The key observation is that IKAROS ChIP-seq peaks, detected with standard PFA fixation, disappear when a second, more stringent crosslinker, DSG, is added. This suggests the binding captured by PFA alone was not stable or specific. The disappearance of these peaks under the more rigorous PFA+DSG condition indicates they were likely weak, transient, or artifactual interactions. Such non-specific binding is not characteristic of functional regulatory elements like active promoters or enhancers but is more likely to occur at random locations across the genome. ANSWER: D |
| Timber-L | The key observation is that ChIP-seq peaks for the transcription factor IKAROS, found using standard PFA fixation, disappeared when the protocol was altered to use PFA+DSG. This suggests the detected protein-DNA interactions are sensitive to the specific crosslinking chemistry. IKAROS is known to functionally bind at active promoters and enhancers to regulate genes. The conclusion is that these disappearing peaks are located precisely at these functional sites. The dynamic chromatin environment at these active regions may be more susceptible to disruption or alteration by the different crosslinking properties of DSG, leading to the loss of the signal. ANSWER: C |
| Timber | The disappearance of IKAROS ChIP-seq peaks when switching from PFA to the more extensive PFA+DSG fixation method indicates that the detection of this binding is highly sensitive to the crosslinking chemistry. IKAROS is a transcription factor known to function at specific, dynamic regulatory regions. The conclusion is that these disappearing peaks are located at its expected sites of action: active promoters and enhancers. These functionally important areas are likely more susceptible to having their epitopes masked or their interactions disrupted by the harsher, more extensive crosslinking of PFA+DSG. ANSWER: C |

Table 10: One case from GPQA-Diamond benchmark. Both Timber-L and Timber reason with mathematical formula while vanilla Insturct model does not.

| | |
|---|---|
| Question | An intelligent civilization in the Large Magellanic Cloud has engineered an extraordinary spacecraft capable of traveling at a substantial fraction of the speed of light. The average lifetime of these aliens is roughly 150 solar years. Now, having Earth as their destination in mind, they are determined to travel with this spacecraft at a constant speed of 0.99999987 c, where c is the speed of light. Approximately, how long will it take for their 22 years old astronaut (from the point of view of the astronaut) to reach the Earth using this incredibly fast spacecraft? A) 81 years B) 72 years C) The astronaut will die before reaching to the Earth. D) 77 years |
| Answer | A |
| Instruct | According to the theory of special relativity, due to time dilation, time passes more slowly for the traveling astronaut than for an observer on Earth. The core of the problem is to determine if the astronaut survives the trip. Time experienced by astronaut: 22 years. Astronaut's initial age: 22 years. Astronaut's age upon arrival: 22 + 22 = 44 years . Average lifespan of the alien species: 150 years. Since the astronaut's age upon arrival (44) is well within the average lifespan (150), the astronaut will survive and reach Earth. This conclusion directly contradicts option (C). Furthermore, the question asks for the travel time from the astronaut's perspective, which is given as 22 years, but this is not an option. The time elapsed on Earth is calculated to be over 43,000 years, also not an option. The premises of the question and the provided answer choices are inconsistent. Based on the physical analysis, the statement in option (C) is factually incorrect. ANSWER: C |
| Timber-L | This problem is solved using the principles of time dilation from special relativity. A critical piece of information, the distance of the journey, is missing. The solution assumes the trip is from the Large Magellanic Cloud to Earth, approximately 163,000 light-years . Calculate Earth Time ($t$): First, we determine the travel time from an Earth observer's perspective. $t = \text{distance/velocity} \approx 163,000 \text{ light-years}/c \approx 163,000$ years. Calculate the Time Dilation Factor ($\gamma$): Next, we find the Lorentz factor for the spacecraft's velocity ($v = 0.99999987c$). $\gamma = 1/\sqrt{1 - v^2/c^2} \approx 1961$. Calculate Astronaut's Time ($t_0$): Finally, we find the time experienced by the astronaut by dividing the Earth time by the dilation factor. $t_0 = t/\gamma \approx 163,000 \text{ years}/1961 \approx 83 years$. The astronaut's age upon arrival would be 22 + 83 = 105 years, which is within their 150-year lifespan, so they survive. The calculated journey time of approximately 83 years is closest to option A. ANSWER: A |
| Timber | This problem is solved using special relativity , specifically length contraction . To find the travel time from the astronaut's point of view, we must first calculate how the distance to Earth appears to shrink for them. The solution assumes a travel distance of 160,000 light-years . Calculate the Lorentz Factor ($\gamma$) : For a speed of $v = 0.99999987c$, the time dilation and length contraction factor is calculated. $\gamma = 1/\sqrt{1 - v^2/c^2} \approx 1961$ Calculate the Contracted Distance ($L'$) : From the astronaut's perspective, the immense distance is contracted by the Lorentz factor. $L' = \frac{\text{Proper Distance}}{\gamma} \approx \frac{160,000 \text{ light-years}}{1961} \approx 81.6$ light-years. Calculate the Astronaut's Travel Time ($t'$) : The time for the astronaut is the contracted distance divided by their velocity (which is nearly the speed of light, $c$). $t' = \frac{L'}{v} \approx \frac{81.6 \text{ light-years}}{c} \approx 81.6$ years The journey takes about 81.6 years for the astronaut. Since they have 128 years of life remaining (150 - 22), they will easily survive. This result is closest to option A. ANSWER: A |

