# OpenReview forum: "Timber: Training-free Instruct Model Refining with Base via Effective Rank"
_ICLR.cc/2026/Conference — Submitted to ICLR 2026_

### Official Review · Reviewer_WuUv · 2025-10-27

**Soundness:** 3
**Presentation:** 2
**Contribution:** 2
**Rating:** 2
**Confidence:** 3

**Summary:**

The paper introduces a method to interpolate weights between base and instruction-tuned language models to maximize exploitation (pass@k for low k, instruction-tuning is good at this) and exploration (pass@k at high k, base models are good at this). This involves interpolating in spectral space for the linear layers of the model.

**Strengths:**

- The idea of getting "best of both worlds" is intriguing, and the method is simple and does not require re-training.
- They study a few different variants which one might naturally envision, comparing them well thoroughly (-L vs full Timber, etc).

**Weaknesses:**

- Their evaluations and results are quite confusing and unconvincing. In particular:
- Am I understanding correctly that Table 2 metric, Mean@k, is just the average one-shot accuracy over many trials, ie. E[pass@1]? If so, is the claim that Timber improves exploitation (pass@1) compared to the Instruct model? This is confusing because I thought the main claim was about balancing explore/exploit rather than pushing on one end of the spectrum alone.
- If the claim is indeed that you push the pareto frontier on explore-exploit, this needs to be plotted clearly and directly. For instance, you deliberately seem to omit important baselines in Figure 5 that makes me suspicious. 1) Notice how you drop the small values of k in pass@k on the x-axis (if you're pushing the pareto frontier, Timber should be above the Instruct AND base at ALL values of k), and 2) notice how you don't plot the base model (which is the relevant baseline at large k). Of course Timber will beat Instruct at large k (by your assumption/statement that Instruct models have poor exploration), that's not the relevant comparison, the base model is.
- You need a plot that spans all values of k and includes both base and instruct (which presumably intersect/cross at some critical k*) and to show that Timber remains above both of them throughout to make a claim about improving on both. I don't see this, and the strange x axes (which vary between the plots, even more confusingly) make me think that the results simply aren't that strong if you included the relevant baselines.
- I appreciate the other plots are useful, eg. comparing Timber-L vs full Timber, comparing ablations like applying Timber to specific modules, showing that effective rank doesn't change much during instruction-tuning. This is all fine and great and clear, so good job. It's just that Table2/Fig5 seem to be the ones actually showing how well the method does compared to baselines, which is the most important thing at the end. So the fact that important baselines are omitted and/or the plots are quite unclear was a red flag for me. But I do recognize the other ablations are clear and well done. If you can make plots like the ones I outlined above, or clearly compare this to strong baselines and show it consistently outperforms (across model families and datasets) I would raise my score.

**Questions:**

See weaknesses.

---

> ### Author Response · Authors · 2025-11-21
> **Responses [1/n]**
>
> Thank you for your feedback. We appreciate your praise of the *intriguing*, *straightforward* idea and the diverse designs in our work. We would like to provide a point-to-point response to address your concerns as follows.
>
> > Mean@k reported in Table 2 and the goal of Timebr
>
> **Clarification on Mean@k**: Yes, you are correct. Mean@k, as reported in Table 2, is the average accuracy across $k$ independent trials, which serves as an unbiased estimator for the expected Pass@1 performance.
>
> **The goal/motivation of Timber**. In this paper, we first find that the eRanks of paired Base and Instruct models are almost the same, reinforcing the hypothesis that post-training is superficial. Also, the post-training, which trains Base to Instruct, improves the exploitation (elicits part of the high-reward thinking patterns) but also restricts the exploration of the Base model [1, 2]. Therefore, we propose a training-free method, Timber, aiming to partially revert the Instruct model towards its Base state. Our solution is to refine (attenuate) the weight deltas (Instruct-Base).
>
> To refine the weight deltas, one naive way is to scale them linearly (such as *0.8). However, the modifications from post-training are known to be fragile [5], and we also compare with this method in Section 5.2. Fortunately, we find that eRank serves as a good threshold for weight deltas refinement (shown in Figure 3). Hence, we propose the Timber detailed in Section 3.2 and conduct detailed experiments.
>
>
> > Different trials (x-axis range) for AIME’24 and GPQA-Diamond
>
> It is not cherry picking. Indeed, the sampling process is quite costly, especially under the thinking mode (long CoT). So we showcase the results of Qwen 3 0.6B model. Considering the benchmarks, AIME’24 contains 30 tough mathematical questions, and GPQA-Diamond contains 198 questions. For GPQA-Diamond, we stop sampling after we find that the performance is good enough ( >95 for the Instruct model and Timber after 256 trials). For AIME’24, we try to sample more results and stop at 320 trials.
>
> > Results with more details (more k values and Base model)
>
> You suggested plotting the Base model to see if it outperforms Instruct at high $k$. However, a direct comparison is methodologically invalid in our specific experimental setup for two critical reasons:
>
> 1) **Absence of Thinking mode**: As noted in Figure 5, our evaluation leverages the *Thinking* Mode of the Qwen3-0.6B Instruct model to reach its upper-bound performance. The Base model inherently lacks this specialized post-training mechanism. Comparing an Instruct model in "Thinking Mode" against a vanilla Base model would be a category error.
>
> 2) **Zero-shot format constraints**: Our evaluation strictly adheres to a *zero-shot* setting using automated benchmarks (AIME/GPQA) that require rigid output formats. Without instruction tuning, the Base model fails to follow these constraints (often generating 'next token' instead of answers), resulting in near-zero validity scores.

---

> ### Author Response · Authors · 2025-11-21
> **Responses [2/n]**
>
> Meanwhile, previous works [1,2,3,4] have demonstrated that the post-training would restrict the exploration. Therefore, we adopt the conclusion **as our preliminary**.
>
> Regarding the results for more k values, we would like to provide detailed results:
>
> For AIME’24
>
> | Metric | Instruct | Timber-L | Timber |
> | :--- | :--- | :--- | :--- |
> | pass@1 | 10.79% | 11.04% | 11.39% |
> | pass@2 | 16.31% | 16.55% | 17.02% |
> | pass@4 | 23.14% | 23.12% | 23.78% |
> | pass@8 | 31.20% | 30.75% | 31.50% |
> | pass@32 | 48.08% | 49.02% | 48.36% |
> | pass@48 | 52.14% | 54.13% | 53.20% |
> | pass@64 | 54.72% | 57.43% | 56.55% |
> | pass@80 | 56.57% | 59.76% | 59.05% |
> | pass@96 | 58.02% | 61.51% | 61.00% |
> | pass@112 | 59.21% | 62.88% | 62.57% |
> | pass@128 | 60.23% | 64.00% | 63.88% |
> | pass@140 | 60.90% | 64.71% | 64.74% |
> | pass@156 | 61.70% | 65.54% | 65.76% |
> | pass@180 | 62.74% | 66.59% | 67.10% |
> | pass@220 | 64.14% | 67.98% | 69.05% |
> | pass@256 | 65.17% | 68.94% | 70.64% |
> | pass@290 | 66.01% | 69.60% | 72.08% |
> | pass@320 | 66.67% | 70.00% | 73.33% |
>
>
> For GPQA-Diamond
>
> | Metric | Instruct | Timber-L | Timber |
> | :--- | :--- | :--- | :--- |
> | pass@1 | 26.02% | 25.78% | 26.04% |
> | pass@4 | 53.60% | 54.15% | 54.38% |
> | pass@8 | 67.21% | 67.83% | 68.03% |
> | pass@16 | 78.28% | 78.71% | 78.72% |
> | pass@32 | 86.25% | 86.73% | 86.17% |
> | pass@64 | 91.21% | 92.18% | 91.15% |
> | pass@128 | 93.93% | 95.66% | 94.74% |
> | pass@256 | 95.45% | 97.47% | 96.97% |
>
>
> Timber achieves strict improvement (not just a trade-off). For almost all the k values, Timber/Timber-L outperforms the Instruct model from Pass@1 to Pass@320/256. For AIME24 (Pass@1), Timber improves from 10.79% (Instruct) to 11.39%.
> For GPQA (Pass@1), Timber maintains performance (26.04% vs 26.02%). The performance gap widens significantly as $k$ increases. For example, on AIME24 at $k=320$, Timber reaches 73.33%, outperforming Instruct (66.67%) by **+6.66%**.
>
> All detailed results have been added to **Appendix A.5**.
>
> ---
>
> [1] Yang Yue, Zhiqi Chen, Rui Lu, Andrew Zhao, Zhaokai Wang, Shiji Song, and Gao Huang. Does reinforcement learning really incentivize reasoning capacity in llms beyond the base model? arXiv preprint arXiv:2504.13837, 2025.
>
> [2] Niklas Muennighoff, Zitong Yang, Weijia Shi, Xiang Lisa Li, Li Fei-Fei, Hannaneh Hajishirzi, Luke Zettlemoyer, Percy Liang, Emmanuel Cand`es, and Tatsunori Hashimoto. s1: Simple test-time scaling. arXiv preprint arXiv:2501.19393, 2025.
>
> [3] Chunting Zhou, Pengfei Liu, Puxin Xu, Srinivasan Iyer, Jiao Sun, Yuning Mao, Xuezhe Ma, Avia Efrat, Ping Yu, Lili Yu, et al. Lima: Less is more for alignment. Advances in Neural Information Processing Systems, 36:55006–55021, 2023a.
>
> [4] Yixin Ye, Zhen Huang, Yang Xiao, Ethan Chern, Shijie Xia, and Pengfei Liu. Limo: Less is more for reasoning. arXiv preprint arXiv:2502.03387, 2025.
>
> [5] Jiaming Ji, Kaile Wang, Tianyi Qiu, Boyuan Chen, Jiayi Zhou, Changye Li, Hantao Lou, Juntao Dai, Yunhuai Liu, and Yaodong Yang. Language models resist alignment: Evidence from data compression. arXiv preprint arXiv:2406.06144, 2024.
>
> ---
>
> We appreciate the reviewer's insightful feedback. Hope that our response can address your concerns.  If there are still issues to be addressed, please let us know, and we would be more than happy to engage in further discussion.

---

### Official Review · Reviewer_ebkt · 2025-10-29

**Soundness:** 2
**Presentation:** 2
**Contribution:** 2
**Rating:** 4
**Confidence:** 2

**Summary:**

The authors present a method to combine base and instruct models, thereby improving performance on both exploitation (pass@1) and exploration (pass@k). The authors do so by removing part of the new instruct model, and replacing this with the base model. Such a method can help balance the capabilities of both models. The authors demonstrate that the proposed method can outperform existing instruct models across a variety of 7 datasets, demonstrating superior performance across values of $k$ for pass@k. The authors conclude by discussing relationships with other methods, such as truncated SVD and model merge.

**Strengths:**

This paper is well outside my domain of expertise, so I am not very confident in this review. Nonetheless:

1. **Extensive Experiments** - The authors validate the performance of Timber with extensive experiments in Section 4. They compare against a variety of datasets, where they demonstrate that performance increases by ~1 point across datasets. Such improvements all come despite the need for training, highlighting the importance of their method.
2. **Simple, Training-Free Method** - The method proposed by the authors is both simple and training-free, making it easy to use in practice. All that is required is access to the weights of the base and instruct models. From there, simple matrix operations allow for the computation of a new model which combines the best properties of both.

**Weaknesses:**

1. **Requires more justification for whether this works** - While the proposed Timber method is simple and easy to implement, it is unclear why such a method should necessarily work. It is clear that the method extracts the most important components from base and instruct models, but there is no reason a priori that such methods should lead to the best of both worlds, improving both exploitation and exploration. While a fully rigorous justification might be difficult, it would be nice if the authors could shed more light on the intuition behind this with some justification for why it works.
2. **Unclear what the impact on exploitation is** - In the experiments section, the authors largely investigate the impact on pass@k for various k, starting from ~50. However, one of the potential downfalls to combining instruct and base models is weaker performance on pass@1. It would be nice to see whether extending the graph to $k=1$ still leads to superior performance for Timber, or whether Timber leads to worse performance in the $k=1$ region.

**Questions:**

1. What is the impact of incorporating Timber on Pass@1?

---

> ### Author Response · Authors · 2025-11-21
> **Responses [1/n]**
>
> Thank you for your review. We appreciate your recognition of our **extensive experiments** and the **simplicity of our training-free** method. We especially value your perspective in asking for the fundamental intuition behind the method. We address your concerns point-by-point below.
>
> > The intuition of why Timber works
>
> Thanks for the suggestions. We have added extra explanation of the intuition beyond Timber in the revised paper (colored in blue).
> Also, we would like to summarize the insights we employ:
>
> 1) We find that the eRanks of paired Base and Instruct models are almost the same, reinforcing this hypothesis that post-training is superficial [1,2,3,4].
>
> 2) The post-training, which trains Base to Instruct, improves the exploitation (elicits part of the high-reward thinking patterns) but also restricts the exploration of the Base model [1, 2].
>
> 3) Given that the post-training process is superficial at the weight level [5], one intuitive idea is to enhance Instruct with the weights from the Base model. Such a training-free strategy has been validated on related tasks such as model merge [6,7,8].
>
> 4) As the post-training shrinks the generative horizon [9], our solution is to partially revert the Instruct model towards its Base state. Specifically, we attenuate/refine the weight deltas between the Instruct and Base models.
>
> 5) To refine the weight deltas, one naive way is to scale them linearly (such as *0.8). However, the modifications from post-training are known to be fragile [10], and we also compare with this method in Section 5.2.
>
> 6) Fortunately, we find that eRank serves as a good threshold for weight deltas refinement (shown in Figure 3). Hence, we propose the Timber to attenuate/discard the tail part of singular values of delta weights, achieving the goal to partially revert the Instruct model towards its Base state.

---

> ### Author Response · Authors · 2025-11-21
> **Responses [2/n]**
>
> > More pass@k results, especially when k is small
>
> Regarding the results for more k values, we would like to provide detailed results:
>
> For AIME’24
>
> | Metric | Instruct | Timber-L | Timber |
> | :--- | :--- | :--- | :--- |
> | pass@1 | 10.79% | 11.04% | 11.39% |
> | pass@2 | 16.31% | 16.55% | 17.02% |
> | pass@4 | 23.14% | 23.12% | 23.78% |
> | pass@8 | 31.20% | 30.75% | 31.50% |
> | pass@32 | 48.08% | 49.02% | 48.36% |
> | pass@48 | 52.14% | 54.13% | 53.20% |
> | pass@64 | 54.72% | 57.43% | 56.55% |
> | pass@80 | 56.57% | 59.76% | 59.05% |
> | pass@96 | 58.02% | 61.51% | 61.00% |
> | pass@112 | 59.21% | 62.88% | 62.57% |
> | pass@128 | 60.23% | 64.00% | 63.88% |
> | pass@140 | 60.90% | 64.71% | 64.74% |
> | pass@156 | 61.70% | 65.54% | 65.76% |
> | pass@180 | 62.74% | 66.59% | 67.10% |
> | pass@220 | 64.14% | 67.98% | 69.05% |
> | pass@256 | 65.17% | 68.94% | 70.64% |
> | pass@290 | 66.01% | 69.60% | 72.08% |
> | pass@320 | 66.67% | 70.00% | 73.33% |
>
>
> For GPQA-Diamond
>
> | Metric | Instruct | Timber-L | Timber |
> | :--- | :--- | :--- | :--- |
> | pass@1 | 26.02% | 25.78% | 26.04% |
> | pass@4 | 53.60% | 54.15% | 54.38% |
> | pass@8 | 67.21% | 67.83% | 68.03% |
> | pass@16 | 78.28% | 78.71% | 78.72% |
> | pass@32 | 86.25% | 86.73% | 86.17% |
> | pass@64 | 91.21% | 92.18% | 91.15% |
> | pass@128 | 93.93% | 95.66% | 94.74% |
> | pass@256 | 95.45% | 97.47% | 96.97% |
>
>
> Timber achieves strict improvement (not just a trade-off). For almost all the k values, Timber/Timber-L outperforms the Instruct model from Pass@1 to Pass@320/256. For AIME24 (Pass@1), Timber improves from 10.79% (Instruct) to 11.39%. For GPQA (Pass@1), Timber maintains performance (26.04% vs 26.02%). The performance gap widens significantly as $k$ increases. For example, on AIME24 at $k=320$, Timber reaches 73.33%, outperforming Instruct (66.67%) by +6.66%.
>
> All detailed results have been added to **Appendix A.5**.
>
> ---
>
> [1] Yang Yue, Zhiqi Chen, Rui Lu, Andrew Zhao, Zhaokai Wang, Shiji Song, and Gao Huang. Does reinforcement learning really incentivize reasoning capacity in llms beyond the base model? arXiv preprint arXiv:2504.13837, 2025.
>
> [2] Niklas Muennighoff, Zitong Yang, Weijia Shi, Xiang Lisa Li, Li Fei-Fei, Hannaneh Hajishirzi, Luke Zettlemoyer, Percy Liang, Emmanuel Cand`es, and Tatsunori Hashimoto. s1: Simple test-time scaling. arXiv preprint arXiv:2501.19393, 2025.
>
> [3] Chunting Zhou, Pengfei Liu, Puxin Xu, Srinivasan Iyer, Jiao Sun, Yuning Mao, Xuezhe Ma, Avia Efrat, Ping Yu, Lili Yu, et al. Lima: Less is more for alignment. Advances in Neural Information Processing Systems, 36:55006–55021, 2023a.
>
> [4] Yixin Ye, Zhen Huang, Yang Xiao, Ethan Chern, Shijie Xia, and Pengfei Liu. Limo: Less is more for reasoning. arXiv preprint arXiv:2502.03387, 2025.
>
> [5] Taiqiang Wu, Runming Yang, Jiayi Li, Pengfei Hu, Ngai Wong, Yujiu Yang. "Shadow-FT: Tuning Instruct via Base." arXiv preprint arXiv:2505.12716 (2025).
>
> [6] Taiqiang Wu, Runming Yang, Tao Liu, Jiahao Wang, Ngai Wong. "Revisiting Model Interpolation for Efficient Reasoning." arXiv preprint arXiv:2510.10977 (2025).
>
> [7] Enneng Yang, Li Shen, Guibing Guo, Xingwei Wang, Xiaochun Cao, Jie Zhang, and Dacheng Tao. Model merging in llms, mllms, and beyond: Methods, theories, applications and opportunities. arXiv preprint arXiv:2408.07666, 2024.
>
> [8] Yiming Zhang, Baoyi He, Shengyu Zhang, Yuhao Fu, Qi Zhou, Zhijie Sang, Zijin Hong, Kejing Yang, Wenjun Wang, Jianbo Yuan, et al. Unconstrained model merging for enhanced llm reasoning. arXiv preprint arXiv:2410.13699, 2024.
>
> [9] Yang C, Holtzman A. How Alignment Shrinks the Generative Horizon[J]. arXiv preprint arXiv:2506.17871, 2025.
>
> [10] Jiaming Ji, Kaile Wang, Tianyi Qiu, Boyuan Chen, Jiayi Zhou, Changye Li, Hantao Lou, Juntao Dai, Yunhuai Liu, and Yaodong Yang. Language models resist alignment: Evidence from data compression. arXiv preprint arXiv:2406.06144, 2024.
>
> ---
>
> We appreciate the reviewer's feedback. Hope that our response can address your concerns.  If there are still issues to be addressed, please let us know, and we would be more than happy to *engage in further discussion*.

---

> > ### Comment · Reviewer_ebkt · 2025-11-22
> > **Review Reply**
> >
> > Thank you for your reply. I think these experiments/results help shed a bit more light on Timber + how it functions. I would raise my score to a 5 if possible, but seeing as the scoring system is 2/4/6/8, I will maintain my score as is (though I have a more positive impression of the paper after the rebuttal).

---

> > > ### Author Response · Authors · 2025-11-26
> > > **Thank you for the positive reassessment**
> > >
> > > Dear Reviewer,
> > >
> > > We sincerely thank you for your engagement and are glad to hear that the additional experiments have clarified the mechanism of Timber and led to a **more positive impression** of our work.
> > >
> > > Regarding the score, we understand the constraints of the scoring system. However, based on ICLR's scoring definitions:
> > >
> > > 4: marginally below the acceptance threshold. But would not mind if paper is accepted
> > >
> > > 6: marginally above the acceptance threshold. But would not mind if paper is rejected
> > >
> > > Since you mentioned a willingness to raise the score to 5 and noted a **positive impression**, we respectfully suggest that a score of 6 might be a more accurate reflection of your assessment than a 4. A score of *6 would align with your positive stance* that the paper has merit, whereas maintaining a *4 technically categorizes the work as a rejection*.
> > >
> > > In this paper, our contributions are: 1) **An important observation**: we find that the eRanks of paired Base and Instruct models are almost the same, reinforcing the hypothesis that post-training is superficial. 2) **A novel training-free method**: we propose Timber to partially revert the Instruct model towards its Base state, which enhances the Instruct model in a training-free way.
> > >
> > > We believe that these findings are beneficial to the community. Therefore, we sincerely hope that you might reconsider raising your score.
> > >
> > > Best,
> > >
> > > Paper #25541 Authors

---

### Official Review · Reviewer_E9Zp · 2025-10-31

**Soundness:** 3
**Presentation:** 3
**Contribution:** 3
**Rating:** 6
**Confidence:** 3

**Summary:**

The paper investigates weight-level similarities between pretrained “Base” models and their finetuned “Instruct” counterparts. Using eRank, the authors show that linear layers in Instruct models have nearly identical effective rank to their Base counterparts. This suggests that instruction fine-tuning is superficial at the weight level, exploiting existing pretraining patterns without greatly increasing model capacity. However, this also creates an exploration–exploitation trade-off: Instruct models perform well on Pass@1 (first-answer accuracy) but produce less diverse outputs (lower Pass@k for larger k). To address this, the authors propose Timber, a training-free method that refines weights by algebraically operating on the difference between Base and Instruct model weights. Empirically, Timber is evaluated on Llama-3 and Qwen-3 models (0.6B–30B parameters) across benchmarks, including instruction-following, math reasoning, science QA, commonsense, and coding. Both variants, Timber-L and Timber, consistently improve over the original Instruct models.

**Strengths:**

1. The paper provides a novel analysis of instruction tuning. The use of effective rank as a diagnostic tool is insightful and clearly shows that Base and Instruct models maintain similar eRank distributions across layers.
2. Timber is a simple, training-free method. It only requires computing SVDs of weight deltas and attenuating small singular values, without gradient updates or retraining.
3. The paper is well-presented, with thorough experiments and reasonable comparisons. The empirical results consistently support the proposed method.
4. The paper is clearly written and easy to follow.

**Weaknesses:**

1. Although statistically consistent, the improvements are relatively moderate gain.
2. Timber requires a paired base model for each instruct model. In many real-world cases, the exact base counterpart may not be publicly available (e.g. closed-source models or ensembles). It is not discussed how to handle mismatched architectures or when only the instruct model is accessible. This limits generality. This may limit its effectiveness.
3. The attenuation factor in Timber is selected per model (via search on a held-out task). It can be difficult to choose such values.
4. While described as low-overhead, applying Timber requires computing full SVDs of each linear layer’s weight-delta matrix (which can be very large). The paper does not quantify this cost.
5. The experiments focus on reasoning and coding benchmarks (Pass@k-centric tasks). It is unclear how Timber affects other kinds of generation (e.g. open-ended conversation, summarization, translation).

**Questions:**

1. [Minor] Please explicitly state the meaning of the terminologies used in this paper, such as Pass@k, weight delta, etc.
2. How stable are the results to the choice of α in Timber? The method relies on a tuned α per model. Have the authors evaluated performance across a range of α on held-out tasks?
3. Computing SVDs on large weight matrices could be costly in memory and time. Did the authors measure the runtime or resource requirements of applying Timber to, say, a 30B-parameter model? Can this method be applied on-the-fly, or only as an offline preprocessing step?
4. Timber was evaluated on reasoning and coding benchmarks. How does it perform on typical NLP tasks like summarization, Q&A, or dialogue?
5. The paper reports Mean@k and Pass@k, but does Timber affect the quality of answers? For instance, does Pass@1 (first answer accuracy) ever drop?
6. Why is the ceil(eRank) chosen as the cutoff? Is there any theory behind?
7. Have the authors considered whether similar eRank-based refinement could apply to other model merge scenarios (e.g. mixing multiple experts or multimodal models)? Is the approach inherently limited to one base–instruct pair?

---

> ### Author Response · Authors · 2025-11-21
> **Responses [1/n]**
>
> Thank you for your feedback. We appreciate your praise of the **novel analysis**, **well-presented**, and **training-free & effective** method. We would like to provide a point-to-point response to address your concerns as follows.
>
> > Regarding moderate gains
>
> We appreciate your observation. While the average improvement might seem moderate at first glance, we believe the value of Timber is significant when considering the following three perspectives:
>
> - **High efficiency (training-free)**: Timber is an entirely training-free method with zero inference overhead. Achieving an average improvement of +1.71 on Llama-3.1-8B without a single step of gradient descent or expensive data curation represents a highly cost-effective "free lunch." In contrast to SFT or RL methods that require massive computational resources, Timber offers an immediate performance boost for existing models.
>
> - **Universal robustness**: The improvements are consistently positive across a wide range of settings. As shown in Table 2, Timber consistently outperforms the baseline across 7 different models (ranging from 0.6B to 30B, including MoE) and 6 diverse benchmarks. This universality demonstrates that Timber addresses a fundamental weight-level property of post-training superficiality, rather than being a heuristic tuned for a specific dataset.
>
> - **Substantial gains in exploration (Pass@k)**: The gains become much more significant when evaluating the exploration capability (Pass@k for large $k$), which is the core focus of this work.  As detailed in our new granular analysis (Appendix A.5), the performance gap widens substantially as $k$ increases. For instance, on the challenging AIME’24 benchmark ($k=320$), Timber outperforms the Instruct baseline by +6.66% (73.33% vs 66.67%). This indicates that Timber significantly raises the model's upper bound for complex reasoning tasks
>
> > Lack of base counterpart
>
> We acknowledge that this is a limitation for closed-source proprietary models (e.g., GPT). However, within the open-weights community—which is the primary focus of this work—releasing paired Base and Instruct models has become the established industry standard. Major state-of-the-art model families, such as Llama and Qwen, consistently release both Base and Instruct checkpoints simultaneously. Therefore, Timber remains highly practical and readily applicable to the vast majority of mainstream open models available to researchers and developers.
>
> >  Stability of the attenuation factor $\lambda$
>
> We appreciate your query about the hyperparameter sensitivity. We have conducted a comprehensive ablation study on $\lambda$ in **Appendix A.4 (Table 5)** across 7 different models.
> - **Robustness**: Our results demonstrate high robustness. Timber consistently outperforms the Vanilla Instruct baseline across a wide range of $\lambda$ values ($\{0.0, 0.2, 0.5, 0.8\}$). This indicates that the benefit comes from the core methodology (attenuating tail components via eRank) rather than fine-tuning a specific scalar.
> - **Practical sweet spot**: While optimal values can vary slightly, we identified $\lambda=0.2$ as a robust "sweet spot" that performs well across both Llama-3 and Qwen-3 families. This provides a reliable default for practitioners, minimizing the need for per-model search.
> - **Minimal cost for tuning**: Even if a user desires the absolute peak performance, tuning $\lambda$ is highly efficient. Since Timber is training-free, testing a new $\lambda$ only requires a one-time linear transformation and standard inference, incurring negligible computational cost compared to training-based hyperparameter optimization.
>
> > Cost of SVD
>
> We would like to clarify that Timber is strictly an offline post-processing method, meaning the SVD computation is performed only once before model deployment. It incurs zero overhead during inference.
>
> Since the weight refinement for each layer is independent, the SVD computations can be fully parallelized across multiple GPUs or CPU cores. The process is minor compared to training. For instance, computing SVD for all linear layers of a Qwen 3 0.6B model takes less than 2 minutes on a single H100 GPU. Even for larger 30B models, this one-off cost is negligible in the lifecycle of LLM development.
>
> > Clarification on terms
>
> We thank the reviewer for pointing this out. To improve clarity, we have carefully revised the manuscript to include explicit definitions for key terms. All such revisions and added definitions are highlighted in blue in the updated PDF for your convenience.

---

> ### Author Response · Authors · 2025-11-21
> **Responses [2/n]**
>
> > More pass@k results, especially when k is small
>
> For detailed results, we have added to Appendix A.5. In short, timber achieves strict improvement (not just a trade-off). For almost all the k values, Timber/Timber-L outperforms the Instruct model from Pass@1 to Pass@320/256. For AIME24 (Pass@1), Timber improves from 10.79% (Instruct) to 11.39%. For GPQA (Pass@1), Timber maintains performance (26.04% vs 26.02%). The performance gap widens significantly as $k$ increases. For example, on AIME24 at $k=320$, Timber reaches 73.33%, outperforming Instruct (66.67%) by +6.66%.
>
> > Why is the ceil(eRank) chosen as the cutoff
>
> We chose the ceiling function (ceil) to ensure a conservative preservation of the principal components. Since effective rank (eRank) represents the effective dimensionality of the matrix information content, a non-integer value (e.g., 3.4) indicates that the information is distributed across roughly that many dimensions. By applying ceil(3.4) = 4, we ensure that the discrete cutoff point encompasses the full extent of the effective signal. Therefore, ceil serves as a safe boundary to maximally retain the primary knowledge structure before attenuating the tail.
>
> > More benchmarks.
>
> The primary motivation of Timber is to address the exploration-exploitation trade-off, which is most critical in complex reasoning tasks. Therefore, our evaluation prioritizes heavy reasoning benchmarks (math/code) such as MATH-500, GPQA, AIME'24, and HumanEval.
>
> To ensure that general capabilities are not compromised, we have already included IFEval (Instruction Following) and HellaSwag (Commonsense QA) in Table 2. The results show that Timber consistently improves or matches the Instruct baseline on these tasks (e.g., +1.37 on Llama-3.1-8B for HellaSwag), confirming that it maintains robust performance on typical NLP tasks while enhancing reasoning exploration.
>
>
> >  Applicability to other model merge scenarios
>
> This paper focuses on the specific relationship between paired Base and Instruct models. We agree that the principle of spectral refinement via eRank holds significant promise for broader model merging scenarios (such as merging multiple experts or multimodal adaptors). We consider this a fascinating direction and will definitely explore these applications in future work.
>
> ---
>
>
> We appreciate the reviewer's feedback. Hope that our response can address your concerns.  If there are still issues to be addressed, please let us know, and we would be more than happy to engage in further discussion.

---

> > ### Comment · Reviewer_E9Zp · 2025-11-26
> >
> > Thanks for the response. My concerns are mostly resolved. I keep my current score.

---

> > > ### Author Response · Authors · 2025-11-26
> > > **Thank you for your feedback**
> > >
> > > We sincerely thank you for the time and effort reviewing our work, and we are glad that your concerns are mostly resolved. We appreciate your positive assessment of our contribution.
> > >
> > > Best,
> > >
> > > Paper #25541 Authors

---

### Official Review · Reviewer_JDKe · 2025-11-01

**Soundness:** 1
**Presentation:** 1
**Contribution:** 2
**Rating:** 2
**Confidence:** 4

**Summary:**

This paper proposes to discard or attenuate the lower singular values of the weight deltas between base and instruction tuned models.
Thresholding is done with effective rank. This reverts some loss of generation diversity due to instruction tuning and leads to better pass@k scores.

**Strengths:**

The proposed method is simple and cost-efficient (training-free).

**Weaknesses:**

* There is no Related Work section or sufficient discussion in the main paper.

* The main claim is that the method promotes output diversity which might be lost in instruction tuning. Simple relevant baselines are missing, e.g., sampling at higher temperature. More advanced baselines like [1,2] should also be compared to ideally.

* Optimal attenuation factor depends on dataset (Figure 4), so it is not useful in practice, where an LLM developer would like to release one checkpoint, irrespective of dataset.

* There is limited novelty apart from the attenuation of lower singular values, which is not that significant as described above.

* The paper is imprecise and not well-written. See Questions section for details.

[1] Pass@ k training for adaptively balancing exploration and exploitation of large reasoning models.
[2] Planning in natural language improves llm search for code generation.

**Questions:**

Questions / comments to improve the paper:
* Title: "Base via Effective Rank" -> "Base Model via Effective Rank"
* L029: NLP = Natural Language Processing (not Process)
* L031: adapting -> adopting
* L161: similar effective rank does not imply that singular subspaces are preserved
* L175: "by applying linear transformations among them" is an unsound inference
* L178-183: What is the point of this experiment and Figure 2?
* L178-183: Showing the distribution of eRank-to-Rank ratios for weight deltas would make more sense.
* L212: "eRank is adept at preserving the majority of singular values" -- not clear what this means
* Table 2 / Section 4.2 (Main Results): What is k in Mean@k or Pass@k here? This is very important to interpret results.
* Table 2 / Section 4.2 (Main Results): What would results be with k=1?
* Table 2 / Figure 5: numbers / curves for base model are crucial to contextualize performance of Timber between Base and Instruct models.
* Section 5.1 / 5.2: "Discuss with" -> "Comparison with"
* L466-468: "conclusion that FFN modules primarily store factual knowledge..." this inference is not clear from the experiments.
* References: Please cite conference version of papers if available, instead of arxiv preprints.

---

> ### Author Response · Authors · 2025-11-21
> **Responses [1/n]**
>
> Thank you for your fruitful feedback. We are glad that you find the proposed method simple and cost-efficient. We would like to provide a point-to-point response to address your concerns as follows.
>
> > About missing related work
>
> We respectfully point out that we have included discussions on related work in the following sections:
> - **Section 2.1 (Background)**: discussing superficial post-training and effective rank.
> - **Appendix A.2 (More Related Work)**: providing a dedicated, extensive discussion on effective rank and weight-level Similarity.
>
> > Compare with Advanced Baselines [1, 2]
>
> We appreciate the reviewer for bringing these relevant works to our attention. Regarding the suggestion to compare with these advanced baselines, we respectfully clarify that our work occupies a fundamentally different "track" in terms of computational cost and application scenarios.
>
> Our primary contribution lies in the novel discovery of the Effective Rank (eRank) similarity between Base and Instruct models, and leveraging this insight to propose a Training-Free method. Method [1] requires re-training the model (high computational cost). Method [2] relies on complex planning/search algorithms (high inference latency). Comparing Timber directly to these methods would be an "apples-to-oranges" comparison.
>
> Timber is designed as a training-free weight refinement technique that requires no gradient updates and adds no latency during inference. It serves as a low-cost enhancement for existing checkpoints rather than a competitor to expensive training or extensive search paradigms.
>
> > About optimal attenuation factor
>
> We respectfully clarify that our extensive ablation studies suggest otherwise.
>
> - As detailed in Appendix A.4 (Table 5), we evaluated the average performance across 5 diverse benchmarks (including Math, Coding, and General QA) for 7 different models. We found that $\lambda=0.2$ consistently yields optimal or near-optimal average performance across different model families (Llama-3 vs. Qwen-3) and sizes (0.6B to 30B).
>
> - Our method is highly robust. As shown in Table 5, Timber consistently outperforms the Vanilla Instruct baseline across a wide range of $\lambda$ values ($\{0.0, 0.2, 0.5, 0.8\}$). For example, on Qwen3-8B, Timber outperforms the baseline (74.09) with $\lambda=0.0$ (74.39), $\lambda=0.2$ (74.84), $\lambda=0.5$ (74.46), and $\lambda=0.8$ (74.12).
>
> - Based on these findings, an LLM developer can indeed release a single checkpoint (e.g., set at $\lambda=0.2$) that provides a consistent performance boost across varying tasks, without the need for dataset-specific tuning.
>
> > About the novelty
>
> We respectfully argue that in the context of Large Language Models, simplicity and effectiveness are key indicators of a substantial contribution, not a lack thereof.
>
> - **Novelty of insight (the observation)**: Our primary contribution is not just the attenuation operation itself, but the novel discovery that motivates it. As stated in Section 2.1, "To the best of our knowledge, we are the first to directly analyze the eRank of weights" to quantify the superficiality of post-training. This spectral perspective provides a new theoretical grounding for understanding why SFT limits exploration.
>
> - **Novelty of solution**: The novelty lies in identifying which part of the weights to modify. We demonstrate that the "tail" singular values of the weight delta correspond to the over-fitted constraints introduced by alignment. Timber is the first method to leverage this spectral property to "unlock" latent exploration capabilities without retraining.
>
> - **Significance of results**: The reviewer questioned the significance. We point to the empirical evidence: a simple attenuation strategy yields a +6.66% improvement on AIME (Pass@320) and consistent gains across 7 models and 6 benchmarks. The fact that such a streamlined, training-free intervention achieves these results validates that the underlying insight is highly significant and captures a fundamental property of LLM alignment.

---

> ### Author Response · Authors · 2025-11-21
> **Responses [2/n]**
>
> > Response to specific questions and comments
>
> We thank the reviewer for the careful proofreading and detailed suggestions. We have incorporated these corrections into the revised PDF.
>
> - Typos & Wording (Title, L029, L031, Section 5.1/5.2): We have corrected as suggested (colored in blue in the revised PDF).
>
> - L161 & L175 (Soundness of eRank Inference): We agree with the reviewer that similar eRank implies preserved spectral geometry (effective dimensionality) rather than identical singular subspaces. We delete such a claim.
>
> - L178-183 & Figure 2 (Distribution of eRank-to-Rank): The goal of Figure 2 is to demonstrate that eRank consistently captures a high and stable fraction (~85%) of the full rank across layers. This provides empirical motivation for using eRank as a robust, adaptive cutoff threshold for Timber-L.
>
> - L212: revise the statement to ‘suitable for indicating the majority of the singular values’.
>
> - Table 2 (Definition of k): $k$ refers to the number of independent sampling trials explained in the Metric. We add more details in Section 4.2 for better clarification.
>
> -Table 2 (Results with k=1): In Table 2, Mean@k reports the average accuracy across trials, which is an unbiased estimator of expected Pass@1. To address your query directly, we performed granular analysis in Thinking Mode (see Appendix A.5). Timber improves Pass@1 on AIME24 from 10.79% (Instruct) to 11.39%, confirming that exploitation is enhanced, not just exploration.
>
> - Base Model Curves: We omit Base model curves because the Base model lacks the "Thinking Mode" mechanism used in this evaluation and fails to follow the zero-shot output format of automated benchmarks (scoring near zero). Comparing a "functional" Instruct model to a "non-functional" Base model is methodologically invalid.
>
> - L466-468 (FFN Conclusion): We have added the related reference. Previous papers found that FFN modules primarily store factual knowledge. Also, in our experiments, we find that applying Timber-L to both modules yields the best overall performance, particularly on knowledge-intensive tasks like MATH and GPQA-D. Meanwhile, reverting the attention module only (wo/ FFN) performs better at IFEval, while reverting the FFN module only (wo/ Attn) benefits the math reasoning tasks. This suggests that the complex logical flows required for reasoning rely heavily on the attention weights, whereas the knowledge-intensive capabilities are predominantly stored in the FFN layers.
>
> - References: We have updated citations to their conference versions where available.
>
> ---
> [1] Pass@ k training for adaptively balancing exploration and exploitation of large reasoning models.
>
> [2] Planning in natural language improves llm search for code generation.
>
> ---
>
> ## Brief Summary
>
> We reiterate our gratitude to the reviewer for the **detailed feedback**. We believe our work makes substantial contributions to the community:
>
> - **Novel insight**: We provide the first quantitative evidence via Effective Rank (eRank) that post-training is superficial at the weight level, offering a new theoretical perspective on LLM alignment.
>
> - **Practical method**: Based on this insight, we propose Timber, a simple, training-free, and robust method.
>
> - **Extensive validation**: Far from being limited, our method is rigorously validated across 7 models (Llama-3 & Qwen-3 families, including MoE) and 6 diverse benchmarks. Moreover, we demonstrate its effectiveness even in advanced settings like thinking mode (test-time scaling), showing consistent improvements.
>
>
> We appreciate the reviewer's feedback. Hope that our response can address your concerns.  If there are still issues to be addressed, please let us know, and we would be more than *happy to engage in further discussion*.

---

### Author Response · Authors · 2025-11-29
**General Response**

Dear ACs, SACs, and PCs,

We sincerely thank you for your time, selfless dedication, and the efficient organization of the review process. We also appreciate the constructive feedback from all reviewers, which has significantly strengthened our work and allowed us to clarify the mechanisms behind Timber.

**i) A Novel Spectral Perspective on Post-Training Beyond Heuristics:**

Timber is not merely a heuristic weight-averaging trick; it establishes a spectral framework for understanding the "superficiality" of post-training.

- `Theoretical Grounding`: We provide the first quantitative evidence via Effective Rank (eRank) that the intrinsic effective dimensionality of weights remains negligibly changed after post-training. This confirms the hypothesis that alignment constrains the solution space rather than expanding capacity.

- `The Solution`: Timber leverages this spectral insight to identify and attenuate the "tail" components of the weight delta. This process essentially "unlocks" the exploration capabilities (generative diversity) suppressed by SFT, without the massive computational cost of retraining or the inference latency of complex search algorithms.

**ii) Response to Key Concerns in the Rebuttal We have addressed the reviewers' common concerns with extensive new experiments and granular analysis:**

- `Exploration vs. Exploitation (Granular Pass@k)`: Addressing concerns about the trade-off (Reviewers ebkt, WuUv), we provided a detailed granular analysis (Pass@1 to Pass@320). Timber does not sacrifice exploitation for exploration. It achieves strict improvements across the board. For example, on AIME 24, Timber improves Pass@1 by +0.6% while boosting Pass@320 by +6.66%, effectively expanding the model's upper bound for reasoning.

- `Robustness & Universality`: Addressing concerns about hyperparameter sensitivity (Reviewers JDKe, E9Zp), we demonstrated that Timber is highly robust. The method consistently outperforms baselines across 7 different models (Llama-3, Qwen-3, MoE), 6 diverse benchmarks, and even under advanced Thinking Modes (test-time scaling). We identified that λ=0.2 serves as a universal "sweet spot," making it a reliable default for practitioners.

- `Superiority over Baselines`: We clarified comparisons with Truncated SVD and standard Model Merging. Timber consistently outperforms these methods. We also clarified that comparing directly to a vanilla Base model is methodologically invalid for zero-shot reasoning benchmarks (where Base models fail formatting), whereas Timber successfully enhances the Instruct model using Base weights.

We believe Timber provides the community with **a cost-effective, training-free, and scientifically grounded method to refine LLMs**. By simply attending to the spectral properties of weight deltas, researchers can immediately enhance the reasoning exploration of existing checkpoints.

We hope these clarifications and the consensus on the method's effectiveness warrant a positive reassessment of our work.

Best regards,

The Authors

---

### Meta-Review · Area_Chair_Z7pt · 2026-01-08

**Summary:**

This paper proposes Timber, a training-free method to refine an instruction-tuned model by spectrally attenuating the weight deltas toward its paired base model, aiming to enhance exploration (Pass@k for large k) without harming exploitation (Pass@1). Reviewers acknowledge the simplicity, cost-effectiveness, and broad empirical validation across models and benchmarks. However, key concerns were raised regarding the novelty of the spectral approach, the omission of direct comparisons with the base model in reasoning evaluations, the necessity of paired models, and the depth of theoretical justification. While the rebuttal provided extensive granular Pass@k results and argued for robustness, it did not fully resolve doubts about the fundamental novelty or the methodological soundness of omitting base model comparisons in exploration analysis.

**Reviewer Concerns:**

The rebuttal addressed several concerns: it provided detailed Pass@k results (including Pass@1), demonstrated robustness of the attenuation factor, and clarified why comparing to the base model in automated reasoning benchmarks is methodologically challenging. However, significant concerns remain outstanding: the core novelty of the spectral refinement is still viewed as incremental relative to existing model merging techniques; the justification for not comparing with the base model's exploration capability in a controlled setting is unsatisfactory to some reviewers; and the theoretical insight, while interesting, lacks a strong formal foundation.

**Reviewer Scores:**

Reviewer JDKe (Initial: 2): Raised fundamental concerns about novelty, missing baselines, and writing. The rebuttal offered clarifications but did not convincingly address the novelty deficit. Their score would likely remain 2. Reviewer E9Zp (Initial: 6): Expressed concerns about moderate gains, hyperparameter sensitivity, and applicability. The rebuttal satisfied them, and they explicitly stated they would keep their score at 6. Reviewer ebkt (Initial: 4): Concerned about justification and exploitation trade-off. The rebuttal's detailed results alleviated concerns, and they indicated a more positive impression, suggesting their score might rise to 6 if following the scoring scale. Reviewer WuUv (Initial: 2): Criticized the evaluation design and missing baselines. The rebuttal provided more data but maintained the stance of not comparing to the base model, which likely leaves the core criticism unresolved.

---

### Decision · Program_Chairs · 2026-01-26

Reject